# Spinopelvic Motion Evaluation in Patients Undergoing Total Hip Arthroplasty and Patient-Specific Target for Acetabular Cup Placement

**DOI:** 10.3390/jpm14121161

**Published:** 2024-12-19

**Authors:** Antonios A. Koutalos, Nifon K. Gkekas, Vasileios Akrivos, Nikolaos Stefanou, Theofilos Karachalios

**Affiliations:** Orthopaedic Department, School of Health Sciences, Faculty of Medicine, University of Thessaly, 41110 Larissa, Greece; nifwn_gekas@hotmail.com (N.K.G.); vasilisakrivos21@gmail.com (V.A.); nikvsteph@gmail.com (N.S.); kar@med.uth.gr (T.K.)

**Keywords:** spinopelvic, sagittal, total hip arthroplasty, reliability, patient specific, safe zone

## Abstract

**Background/Objectives**: Instability is a major reason for revision after total hip arthroplasty (THA), and acetabular cup placement in the “traditional” safe zone does not protect against dislocations. Spinopelvic mobility may play a role in impingement and dislocation after THA. Personalized acetabular cup placement that incorporates spinopelvic mobility is currently lacking in the literature. **Methods**: The spinopelvic motion of 116 patients was evaluated during preoperative planning. All patients underwent radiological assessments with an anteroposterior pelvis radiograph in the standing and supine positions and a lateral view of the lumbar spine and pelvis in the standing and sitting positions. The pelvic incidence, pelvic tilt, sacral slope, standing anterior pelvic plane tilt, sitting anterior pelvic plane tilt, and lumbar lordosis angle were measured, and the degree of pelvic motion from standing to sitting was calculated. The development of the patient-specific target for the acetabular cup was based on the mean mobility of the whole group and the specific posture of each patient. **Results**: The average pelvic incidence was 51.0 ± 13.1 degrees, the sacral slope was 35.0 ± 10.3 degrees, the pelvic tilt was 16.0 ± 13.3 degrees, the standing anterior pelvic plane tilt was 3.4 ± 12 degrees backward, and the degree of lumbar lordosis was 39.5 ± 11.3 degrees. The mean spinopelvic mobility was 27.3 ± 13.4 degrees. The measurements had good to excellent interobserver and intraobserver reliability. On the basis of these measurements, we developed a novel algorithm for a patient-specific target for acetabular cup placement. **Conclusions**: the evaluation of spinopelvic mobility has good to excellent interobserver and intraobserver reliability and can be used for personalized acetabular cup placement.

## 1. Introduction

Total hip arthroplasty (THA) is considered the most successful management option for end-stage arthritis of the hip. Even though the revision rate is acceptable, dislocation may occur either acute or in the long term. Dislocation of THA is associated with pain, lower functional outcomes, and increased cost in case of revision [1].

The dislocation rate varies from 0.2% to 10% in primary THAs. Risk factors include head size, surgeon experience, obesity, and age [2]. Recently, interest in the spinopelvic relationship and its implication for impingement and dislocation have increased among many research groups [3,4,5,6]. Backward pelvic motion while sitting is unique for every patient, and it is associated with acetabular opening or anteversion when sitting. It has been suggested that placing the acetabular cup in the Lewinnek “safe zone” [7] does not protect from dislocation; instead, a patient-specific safe zone has been proposed [8,9,10]. In the hip–spine classification system, patients can be categorized according to sagittal spine balance and their spinopelvic motion [9]. Patients can have a balanced (category 1) or unbalanced spine (category 2) and can have a normal or a stiff spine. For each category, a range of values for acetabular cup anteversion is proposed, or the use of dual mobility cups is advised. For example, patients with stiff spines should have their acetabular cups placed more anteverted (15 to 20 degrees), while in patients with stiff and unbalanced spines the use of dual mobility cups should be considered [9].

In the literature, however, either a range of values for cup placement is given or the spine stiffness is not “quantified” in a patient-specific way [8,10]. In the study by Grammatopoulos et al., a target for acetabular cup placement is proposed, but this method relies on graphs which are difficult to use in daily practice [8]. On the other hand, Wiznia et al. suggested a stepwise approach to finding the most appropriate acetabular cup position according to the patient’s spinopelvic mobility, but they did not suggest a specific target because they did not quantify the spine stiffness [10]. In addition, with the increased use of navigation and robotics in THA, a specific target of anteversion and inclination is preferred. All these methods rely on measurement of spinopelvic motion in the sagittal plane during preoperative planning. Subsequently, the reliability of these evaluations is important.

The primary goal of this study was to evaluate the major characteristics of the spinopelvic mobility in patients undergoing primary THA and define the reliability of these measurements. The secondary aim was to incorporate patients’ spinopelvic mobility in a personalized target for the acetabular cup position.

## 2. Materials and Methods

This was an observational cohort study conducted in University General Hospital of Larissa, a tertiary teaching hospital. From January 2022 to December 2022, 127 consecutive patients who were scheduled for primary total hip arthroplasty underwent a dedicated radiographic examination. The exclusion criteria were developmental dysplasia of the hip types B (low dislocation) and C (high dislocation) according to Hartofilakidis [11] and inability of the patient to sit in a chair making a right angle between the thigh and the trunk due to severe hip contracture or any other reason. Poor-quality radiographs that were not diagnostic or missing the points of interest were discarded. Patient demographics and the reason for THA were recorded. This study was approved by the Ethics Committee of the University General Hospital of Larissa (No. 43445/10-01-2022).

All eligible patients underwent a series of standardized radiographs. These included an anteroposterior (AP) pelvis radiograph in the standing and supine positions and a lateral view of the lumbar spine and pelvis in the standing and sitting positions. The lateral view in the sitting position was performed with the patient sitting on a chair comfortably making a right angle between the thighs and the trunk. All radiographers were trained by two orthopedic surgeons (A.K., N.S.) regarding the position of the patients during the examination and supervised the first ten cases. AP pelvis radiographs were used for preoperative planning which included sizing, leg length discrepancy, and femoral neck osteotomy. In addition, forward or backward rotation of the pelvis was noted if the AP standing pelvis radiograph resembled an outlet or an inlet view, respectively. The two lateral radiographs were then used for the sagittal evaluation of the spinopelvic unit.

The assessment of the sagittal plane included measurements of the pelvic incidence (PI), sacral slope (SS), and pelvic tilt (PT), which are linked together by the equation PI = SS + PT [12]. Furthermore, the anterior pelvic plane tilt (APPt) in the standing and sitting positions was measured. Finally, the degree of lumbar lordosis (LL) was recorded (Figure 1; Figure 2). All assessments were performed by a fellowship-trained adult reconstruction consultant (A.K.) and a fourth-year resident (N.G.). These researchers re-reviewed the same images after one month. All measurements, on both occasions, were used for calculating average values and measuring interobserver and intraobserver reliability.

Normal spinopelvic mobility was defined as the mean spinopelvic motion ± 1 SD. The spinopelvic unit was characterized as stiff when the degree of pelvic motion was less than the mean − 1 SD, whereas the hypermobile unit was defined when the degree of motion was greater than the mean + 1 SD [4].

The development of the patient-specific target for the acetabular cup was based on the mobility of the whole group and the specific posture of each patient. The goal was to place the acetabular cup in the position of 40 degrees of inclination and 20 degrees of anteversion when the patient was standing [13]. This process included three steps. The initial target of 40/20 was adjusted for the APPt of each patient. For every degree of forward or backward rotation of the APPt, the anteversion and the inclination were increased or decreased by 0.7 and 0.3, respectively [14,15]. The second adjustment was driven by the stiffness of the spinopelvic unit. Given the fact that the mean mobility of the whole group is known, we added 0.7 degrees of anteversion and 0.3 degrees of inclination for every degree of “stiffness” beyond the mean. When the patient’s spinopelvic mobility was greater than the mean, the anteversion angle and the inclination angle were reduced, respectively. This represents a novel method that allowed us to calculate a patient-specific target for acetabular cup placement. Finally, if the spine was unbalanced, meaning that there was a mismatch between the lumbar spine and the pelvic incidence greater than 10 degrees (PI–LL > 10), a suggestion was made for using dual mobility, big femoral head, or high offset. Thus, the final inclination and anteversion of the acetabular cup was corrected to the following:A = 20 + 0.7 × APPt_stand_ + 0.7 × (ΔAPPt_mean_ − |ΔAPPt|)
I = 40 + 0.3 × APPt_stand_ + 0.3 × (ΔAPPt_mean_ − |ΔAPPt|)
where ΔAPPt_mean_ is the mean spinopelvic mobility of the whole study group, and ΔAPPt is the difference in APPt between the standing and the sitting positions of a specific patient.

All surgeries were performed through posterior approach by three fellowship-trained adult reconstruction surgeons. Patient positioning in lateral decubitus position included two supports in the front to stabilize anterior superior iliac spines and one at the back to support in the sacrum, so that the pelvis was parallel to the walls and the floor, avoiding any rotation or cephalad/caudal inclination. The acetabulum was prepared with sequential reamers avoiding medialization in order to reproduce the center of rotation of the patient. A press fit cup was implanted with the aid of a mechanical alignment guide and a digital goniometer trying to place the cup at the planned angles of anteversion and inclination. Navigation or robotic system was not used. Complications were recorded at follow-up visits at 30 and 90 days. Any dislocations were documented from emergency visits records.

Appropriate descriptive statistics were used for demographic data. Agreement between the raters was evaluated with the interclass correlation coefficient (ICC). For intraobserver ICCs, a two-way mixed model based on absolute agreement and a single rater was utilized, whereas for interobserver ICCs, a two-way mixed model based on absolute agreement of multiple raters was selected. It was calculated that 102 patients would be needed to have an ICC above 0.75, with an acceptable power above 80%. ICC values below 0.5 are considered to represent poor reliability, values between 0.5 and 0.74 are considered moderate, values between 0.75 and 0.9 are considered good, and finally, values above 0.90 are considered excellent reliability. Significance was set to *p* < 0.05.

## 3. Results

A total of 127 patients were scheduled for primary THA during the study period. After applying the exclusion criteria, 116 patients remained for assessment. Four patients were excluded because of type B or C DDH, three patients were excluded because of inability to undergo the radiographic examination due to severe hip contracture, and four patients were excluded because of poor quality radiographs. In total, 73% of the patients (85/116) were women, and 80% (93/116) had osteoarthritis. No early dislocations were noted. One patient was diagnosed with deep vein thrombosis, and one was diagnosed with a urinary tract infection. Finally, one patient developed a superficial infection that was successfully managed with antibiotics.

The mean spinopelvic mobility (ΔAPPt_mean_) was 27.3 ± 13.4 degrees based on all measurements. The mean PI was 51.0 ± 13.1 degrees, the SS was 35.0 ± 10.3 degrees, the PT was 16.0 ± 13.3 degrees, the standing APPt was 3.4 ± 12 degrees backward, and the LL was 39.5 ± 11.3 degrees (Table 1).

Intraobserver reliability was estimated to be good to excellent for the majority of the measurements of both raters (Table 2). The best intraobserver ICC values, regarding the consultant, were recorded for PT and APPt_stand_ measurements, whereas the best ICC values regarding the resident were noted for PT and PI. Interobserver ICC measurements are depicted in Table 3. PI, PT, and LL had excellent reliability, while SS, APPt_stand_, and APPt_sit_ showed good reliability.

In the seated position, the APPt further decreased. In total, 20% of the patients had stiff spines, while 14% were hypermobile. The target for anteversion and inclination was thus calculated as follows (Figure 3, Figure 4 and Appendix A):A = 20 + 0.7 × APPt_stand_ + 0.7 × (27.3 − |ΔAPPt|)
I = 40 + 0.3 × APPt_stand_ + 0.3 × (27.3 − |ΔAPPt|)

## 4. Discussion

In this study, we estimated that the mobility of the spinopelvic unit of patients who are scheduled for primary THA is 27.3 ± 13.4 degrees. Measurements of spinopelvic radiographic parameters in the sagittal view have good to excellent reliability. On this basis, we calculated a personalized target of acetabular cup position, so that the patient has the optimal placement of the cup in the standing position. Furthermore, this optimal position takes into account the mobility of the pelvis in order to prevent impingement and dislocation.

Interobserver reliability was good to excellent for PI and LL, whereas it was moderate to good for SS, APPt_stand_, and APPt_sit_. Better intraobserver values were recorded, but again, APPt_stand_ had moderate to excellent reliability for the resident, and APPt_sit_ had moderate to excellent reliability for the consultant. The points of interest that were most difficult to identify were the pubic symphysis and the anterior superior iliac spines, which were used to calculate the APPt_stand_ and APPt_sit_. These measurements had the lowest interobserver and intraobserver values, as mentioned above. The use of EOS imaging may be a better option for radiological evaluation and would have increased the interobserver and intraobserver reliability, but the EOS system is unavailable in our setting [16]. Nevertheless, good ICCs were noted with the utilization of routine radiographs. In the future, artificial intelligence may provide surgeons with automated calculations and recommendations based on the individual spinopelvic anatomy and motion of each patient [17].

The reliability of spinopelvic sagittal alignment measurements has been studied by other authors as well [18]. Pernaa et al. found excellent intraobserver reliability for PI, PT, SS, and LL but only good to excellent interobserver reliability for the same measurements. However, the authors concluded that these measurements can be safely used in research and clinical practice [18]. Similarly, McIntosh et al. reported excellent interrater reliability for PI-LL measurements and good reliability for SS measurements [19]. This finding is consistent with our results. In another study, Kleeman-Forsthuber et al. reported good interobserver agreement for APPt_stand_, as we found in our study [20]. In contrast, he reported excellent interobserver reliability of SS measurements, with an ICC value of 0.926. In this study, good reliability was noted for the measurement of SS with an ICC value of 0.83. The intraobserver ICC values for APPt_stand_ in the study by Kleeman-Forsthuber et al. were lower and ranged from 0.637 to 0.862, whereas in this study, good to excellent agreement was noted, with ICC values of 0.99 for the consultant and 0.82 for the resident. The familiarity with preoperative planning in the sagittal plane of the raters in the present study may be associated with better interobserver and intraobserver ICC values. Other studies with experienced raters reported good to excellent reliability over the same measurements [21,22]. Finally, Kleeman-Forsthuber et al. reported that the interobserver agreement of the ante-inclination measurement was moderate. We did not measure this parameter in the sagittal views as this was not necessary for the construction of the equations of personalized acetabular cup anteversion and inclination. Therefore, for the formation of equations, we relied only on parameters with good or excellent measurement reliability.

The measurement of spine mobility has also been investigated by other researchers. Kawanabe et al. studied a group of patients undergoing THA and reported that the ΔAPPt_mean_ was 28.7 ± 8.6 degrees, which is very similar to our results of 27.3 ± 13.4 degrees [5]. Nishihara et al. reported that the degree of pelvic extension from standing to sitting was 32 degrees, but this was a simulated study with CT scans, without direct measurements of APPt_stand_ and APPt_sit_ from lateral views [23].

The implantation of the acetabular cup in a specific target requires surgical experience and a good view of the acetabulum. Approach does not seem to affect the accuracy of cup positioning [24], but level of training does [25]. However, the use of navigation or robotics increases accuracy in acetabular cup placement regardless of experience or approach [26,27]. If satisfactory precision in cup placement can be achieved, the ultimate goal of using this algorithm in daily clinical practice would be a reduction in dislocations and probably elimination of impingement. In addition, placement of the acetabular cup in a personalized target might increase the range of motion and improve the functional outcome of patients. However, these postulates need to be validated in future studies. 

There are many attempts to define a patient-specific approach for implant positioning during THA in the literature. Wiznia et al. adjusted the anteversion of the cup due to APPt in the standing position and stiffness of the spine [8]. However, the adjustment for the stiffness was not “quantified” as in our method. The authors increased the anteversion of the acetabular cup by 10 degrees when stiffness was noticed, which was defined as the difference in SS between standing and sitting being less than 10 degrees. Furthermore, Grammatopoulos et al. reported a method for defining the optimal cup position based on the sagittal plane preoperative planning. The authors used the combined sagittal index (SCI) to find the best ante-inclination angle and then, based on nomograms, determined the anteversion and inclination angles on the coronal plane [6]. However, this method is rather complicated and not familiar to surgeons who are used to planning in the coronal plane. A similar methodology was followed by Bodner, who calculated the anatomic target ante-inclination for the cup in the sagittal plane. This algorithm takes into consideration both the native pelvic tilt in the standing position and the stiffness of the spine as in the present study [28]. However, we believe that the novel algorithm presented in this study is simple enough yet can provide the surgeon with a patient-specific target for each patient.

The limitations of this study include a lack of clinical data. Firstly, there are no clinical data indicating that when the cup is placed in the suggested position, there will be fewer dislocations. However, the relationship between spinopelvic mobility and hip instability has been shown for late dislocations [29]. Furthermore, a recent observational study reported a low rate of dislocations when the spinopelvic mobility was taken into account for the acetabular cup placement [30]. This method is useful for, but not limited to, surgeons with a preference for the posterior approach. Secondly, this study was based on a specific cohort in a single center. One might argue that the results of our study could not be generalized in other populations, but the algorithm enables adaptation to other populations as well if the specific mean spinopelvic mobility of the population (ΔAPPt_mean_) is known. In addition, we did not evaluate the position of the cup after surgery, as the utilization of this novel algorithm has meaning when navigation or robotic systems are available. A comparative study encompassing navigation during placement of the acetabular cup is underway to determine the feasibility of placing the acetabular cup in the recommended position and the impact of this method on clinical outcomes which include pain, range of motion, revision rate for dislocation or any reason, and hip forgotten joint score. Finally, more raters could have produced more robust results.

## 5. Conclusions

In conclusion, there is good to excellent reliability in the evaluation of spinopelvic motion. A personalized target for acetabular cup placement on the basis of spinopelvic motion of patients undergoing THAs can be defined with an easy method. The algorithm can be incorporated into the software of navigation or robotic systems, which are crucial for the accurate implantation of an acetabular cup. The effect of this algorithm on dislocation rate, impingement, or functional outcome needs to be clarified in future studies.

## Figures and Tables

**Figure 1 jpm-14-01161-f001:**
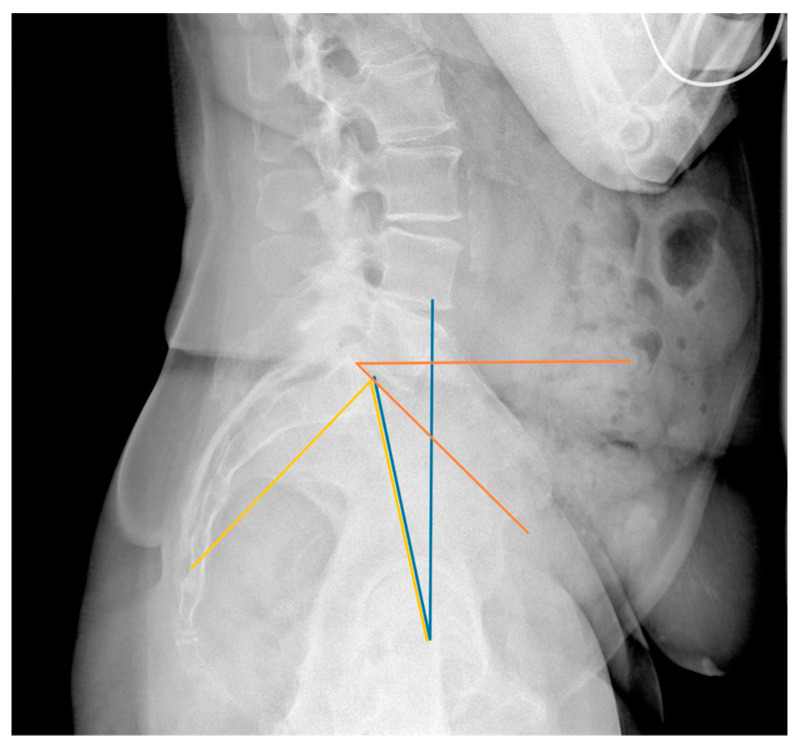
Spinopelvic measurements. Pelvic incidence (PI), represented by the yellow line, is the angle between the line perpendicular to the middle of the sacral end plate and the line connecting this middle of the sacral end plate and the center of the femoral heads. Pelvic tilt (PT), depicted by the blue line, is the angle between the line connecting the center of the femoral heads and the midpoint of the sacral end plate and the vertical axis. Finally, sacral slope (SS), illustrated by the orange line, is the angle between the sacral end plate and the horizontal axis. PI is equal to the sum of the PT and the SS (PI = PT + SS).

**Figure 2 jpm-14-01161-f002:**
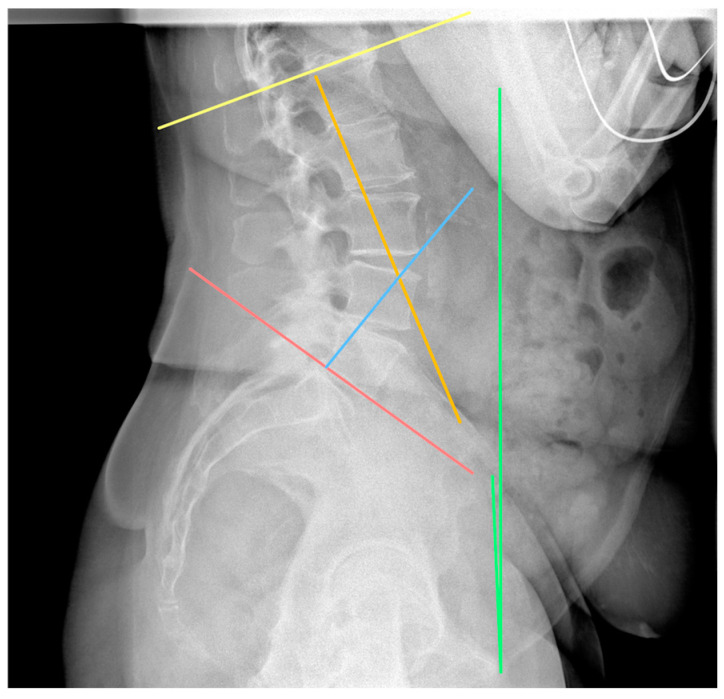
Lumbar lordosis (LL) and anterior pelvic plane tilt (APPt). LL, which is depicted as the angle between the orange and the blue lines, is defined as the Cobb angle formed between the lower-end plate of the fifth lumbar vertebra and the upper-end plate of the first lumbar vertebra. APPt, which is represented by the green line, is the angle formed by the line connecting the pubis symphysis and the anterior superior iliac spines and the vertical axis. In this figure, the APPt in the standing position is shown (APTT_stand_). Similar angle can be measured in the sitting position.

**Figure 3 jpm-14-01161-f003:**
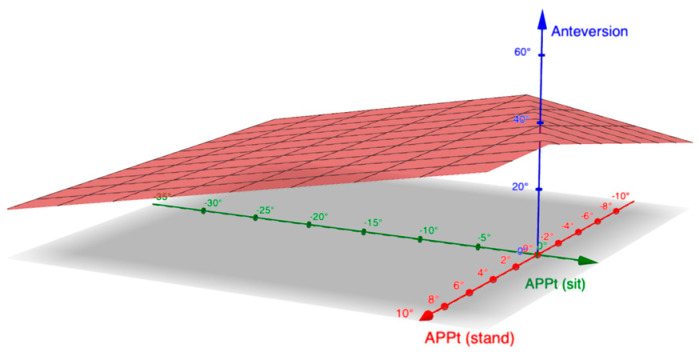
Graph depicting the target anteversion in relation to anterior pelvic plane tilt (APPt) in standing and sitting positions. X−axis represents APPt in sitting position (APPt_sit_), y−axis represents APPt in standing position (APPt_stand_), and z−axis is the suggested anteversion according to the algorithm.

**Figure 4 jpm-14-01161-f004:**
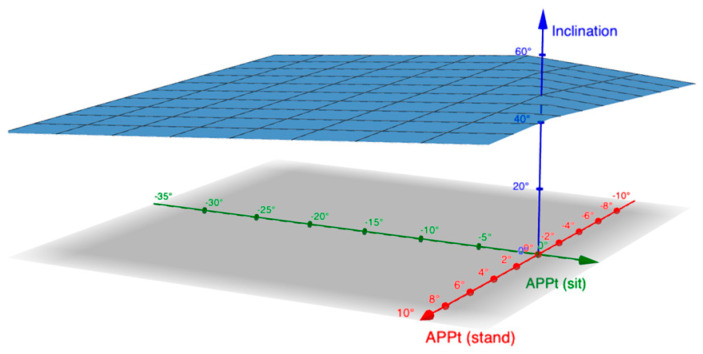
Graph depicting the target inclination in relation to anterior pelvic plane tilt (APPt) in standing and sitting positions. X−axis represents APPt in sitting position (APPt_sit_), y−axis represents APPt in standing position (APPt_stand_), and z−axis is the suggested inclination according to the algorithm.

**Table 1 jpm-14-01161-t001:** Radiographic assessment. Means and standard deviation based on all measurements.

	Mean	SD
PI	51.0	13.1
SS	35.0	10.3
PT	16.0	13.3
APPtstand	−3.4	12.0
APPtsit	−30.7	15.3
ΔAPPt (APPtstand − APPtsit)	27.3	13.4
LL	39.5	11.3

**Table 2 jpm-14-01161-t002:** Intraobserver measurement. ICC for study measurements.

	PI	SS	PT	APPt_stand_	APPt_sit_	LL
Rater 1 (consultant) (ICC value)	0.91	0.83	0.98	0.99	0.82	0.94
Rater 1 (consultant) 95% CI	0.82–0.96	0.67–0.92	0.95–0.99	0.97–0.99	0.63–0.92	0.89–0.97
Rater 2 (resident) (ICC value)	0.96	0.95	0.98	0.82	0.93	0.91
Rater 2 (resident) 95% CI	0.93–0.98	0.90–0.97	0.95–0.99	0.65–0.92	0.85–0.97	0.80–0.92

**Table 3 jpm-14-01161-t003:** Interobserver measurements.

	PI	SS	PT	APPt_stand_	APPt_sit_	LL
ICC value	0.91	0.83	0.98	0.99	0.82	0.94
95% CI	0.82–0.96	0.67–0.92	0.95–0.99	0.97–0.99	0.63–0.92	0.89–0.97

## Data Availability

Data are available upon reasonable request.

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
