# Peer review of "Spinopelvic Motion Evaluation in Patients Undergoing Total Hip Arthroplasty and Patient-Specific Target for Acetabular Cup Placement"

_jpm, 2024, doi:10.3390/jpm14121161_

Round 1

Reviewer 1 Report

Comments and Suggestions for Authors

The study is good but requires major corrections.

1. Abstract 

a) It is better to include one sentence that states the gap in research or drawbacks from previous studies.

b)"...developed a novel algorithm..." it is not clearly explained in the method.

2. Introduction

a) 1st para - it is too short and not even significance to the whole section.

b) Research gap is unclear. Better to include drawbacks from previous literature where the gap can be identified from this.

c) Too short. Research background should be highlighted more in this section.

d) Issue or drawbacks of spinopelvic motion should be highlighted in this section since the conclusion and title are reflecting to this issue.

3. Methodology

a) Hospital - which? not clear.

b) Ethical approval and consent form should be included in the appendix section.

c) Total number of patients should be clearly stated in this section.

d) Patient screening should be clearly stated. How authors select and reject?

e) Surgery procedures are not clear. Better to be specific and detail.

f) How do the authors record the complications and dislocations? It's not clear as well.

g) Statistical analysis was not clear at this stage.

4. Results

a) It is suggested to include more interesting results, ie: bar chart, graph etc.

b) Some results should be in qualitative data, ie: x-ray images

5. Discussion

a) Comparison with other literature - very limited at this stage. I suggest to include more.

b) Limitation of study should be included.

6. References

a) Better to add more and cite recent literature. Most of them are not recent.

Comments on the Quality of English Language

Minor grammatical errors found

Author Response

Author's reply was uploaded as a pdf document

Reviewer 2 Report

Comments and Suggestions for Authors

Spinopelvic motion evaluation and patient specific target for acetabular cup placement

REVIEW

Overall Recommendation

The article provides valuable insight into personalized acetabular cup placement in total hip arthroplasties, with a novel algorithm that has the potential to impact clinical practice. However, before publication, the following improvements are recommended:

  1. Expand the introduction with a more detailed literature review and international context.
  2. Address potential external validity concerns in the methods section.
  3. Provide more visual representation of the results and expand the discussion of clinical implications and future research. Firstly, how this algorithm influence their work in the studied group, and how you can generalize it to an international level!
  4. Strengthen the conclusion to focus on the practical impacts of the findings.

Title

·         Positive: concise and informative

·         Negative: It would be better also to specify the target population

Abstract:

·         Positive: Well structured and it is a good summary of the background/objectives, methodology, key findings and conclusion.

Introduction

  • Positive: It presents the limitations of the "traditional" safe zone method and justifies the study's focus on a patient-specific approach.
  • Negative: It should be extended and presents more details about the existing studies on spinopelvic motion, outilining the ongoing debates on the field. Also, using a specific cohort (Greek population and one orthopedic department), could be elaborated to clarify its relevance to international population.

Methods

  • Positive: The methodology is comprehensive and  development of the personalized algorithm is one of the key innovations of the study, and its step-by-step explanation is logically presented.
  • Negative: The study lacks information on the external validity of the method beyond the studied population. The algorithm is promising, but a comparative analysis with other existing algorithms for acetabular cup placement would add validity to the study. Additionally, more detail on how surgeon experience, posterior approach or potential variability in radiographic interpretation might affect the results would be useful

Results

  • Positive: The results are well-structured, and the statistical presentation, including tables are a strong point of the study, giving reliability, and enhances the credibility of the findings.
  • Negative: Lack of visual aids limits the interpretability of the data. The authors could add graphs and charts to present more clear the relationship between spinopelvic parameters and the adjusted acetabular cup placements. Also for a better clarity, x-ray pre and postoperative should be used and to understand how this algorithm lower the dislocation rate or influence the overall results of the THA in their patients.

Discussion

  • Positive: The discussion provides a balanced interpretation of the findings, reinforcing the validity of spinopelvic mobility in optimizing acetabular cup placement. The authors acknowledge the limitations of current findings.
  • Negative: The discussions should be more focus on how this work will influence the clinical practice and will change the rate of dislocation on an international scale. In my opinion they were not able to show this in their cohort population and is quite hard to understand how their results could influence our clinical practice. Also, there is limited discussion of future work or clinical validation studies that could confirm whether this algorithm indeed leads to fewer complications (e.g., dislocations, revision surgeries) in the long term.

Conclusion

  • Positive: The conclusion is succinct and reiterates the reliability of spinopelvic evaluations in the study. The idea of using a personalized target for acetabular cup placement is innovative and has significant potential.
  • Negative: The conclusion could better emphasize how this study advances the field and suggest specific next steps for clinical or research follow-up. Mentioning the potential integration of navigation or robotic surgery into the algorithm is intriguing but requires further elaboration on how this could be achieved.

References:

  • Positive: is adequate
  • Negative: It should be strengthened with more recent publications, especially on artificial intelligence and imaging technologies in THA.

Author Response

(The authors gave the same response as above.)

Round 2

Reviewer 1 Report

Comments and Suggestions for Authors

The authors have carefully addressed all the comments. The contents are now ready for publication.